# BPL: Generalizable Deepfake Detection via Bias-only Pair-aware Learning

Yuxiang Xu[1]  Rundong He[2]  Zhiyuan Yan[3]  Yicong Dong[1]  Zhongyi Han[1]  Xiaoyan Wang[4]  Yilong Yin[1]

## Abstract

The detection of synthetic images has traditionally been framed as a binary classification problem. However, we argue that this formulation overlooks a fundamental structural property of generative datasets: **synthetic images are not independent samples, but are implicitly paired with real images sharing the same semantic source**. Existing methods treat real and fake images as independent instances, failing to capture generation-induced relational discrepancies in real–fake pairs. Moreover, models tend to rapidly overfit to seen fake patterns, leading to poor generalization to unseen ones. To overcome these challenges, we propose a novel detection framework that explicitly mines real–fake pairs by constructing source-guided mappings or leveraging nearest-neighbor relationships in the CLIP embedding space. We then introduce pair-wise discrepancy learning that explicitly enlarges generation-induced deviations and discrepancy inversion to mitigate overfitting. Moreover, to preserve pretrained semantic representations while improving generalization, we adopt a bias-only fine-tuning scheme that restricts model capacity during adaptation. Extensive experiments show that our approach achieves superior generalization across unseen fake patterns.

## 1. Introduction

Driven by rapid advances in generative models, from Generative Adversarial Networks (GANs) (Goodfellow et al., 2014) to diffusion models (Rombach et al., 2021; Razzhigaev et al., 2023; Rombach et al., 2022), generative AI is now capable of synthesizing images with near-photorealistic

[1]School of Software, Shandong University, Jinan, China [2]Department of Computing, The Hong Kong Polytechnic University, Hong Kong, China [3]Peking University Shenzhen Graduate School, Shenzhen, China [4]Information Technology Service Center of People's Court, Beijing, China. Correspondence to: Rundong He <herundong0@gmail.com>.

*Proceedings of the 43rd International Conference on Machine Learning*, Seoul, South Korea. PMLR 306, 2026. Copyright 2026 by the author(s).

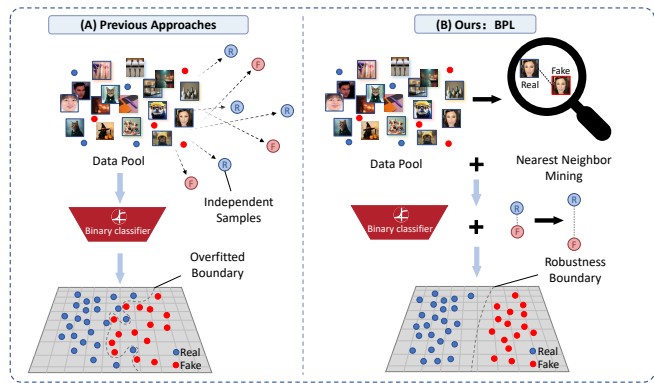

*Figure 1.* Conceptual comparison between previous methods and our proposed BPL framework.

fidelity, often indistinguishable from real ones to both humans and conventional detectors (He et al., 2016; Qian et al., 2020b). While these developments significantly lower the barrier to content creation, they also raise serious concerns regarding misinformation, copyright infringement, and malicious applications such as deepfakes (Korshunov & Marcel, 2018; Thies et al., 2016). Consequently, developing reliable and robust detection methods for AI-generated content has become critical to safeguarding media integrity and maintaining public trust.

Early research on AI-generated image (AIGI) detection (Rössler et al., 2019; Wang et al., 2020) primarily focuses on exploiting model-specific fingerprints and low-level artifacts left by generative processes. Representative approaches include frequency-domain analyses (Ding et al., 2025; Qian et al., 2020a), spatial or textural feature modeling (Wang et al., 2020), and reconstruction-based methods (Wang et al., 2023; Ricker et al., 2024). While effective on in-domain benchmarks, these methods often depend heavily on generator-specific patterns or high-frequency artifacts, leading to pronounced performance degradation when faced with unseen fake patterns.

Recent studies incorporate Vision-Language Models (VLMs) to alleviate the generalization bottleneck by relying on its rich representational capacity (Radford et al., 2021). Representative approaches include semantic consistency modeling based on VLMs (Yan et al., 2025a) and image-text alignment–based discrimination methods (Liu

et al., 2025). Benefiting from the strong and bias-resilient representations of vision-language models such as CLIP, these methods generalize better to unseen generators and demonstrate increased robustness compared to approaches relying on low-level artifacts.

Most existing approaches formulate synthetic image detection as a binary classification problem, where real and fake images are treated as independent instances. However, this formulation overlooks a fundamental structural property of generative data: **synthetic images are not independent samples, but are implicitly paired with real images sharing the same semantic source**. Since ignoring such real–fake relationships, current detectors fail to capture generation-induced relational discrepancies that naturally arise between paired samples. Moreover, models often rely on generator-specific cues and rapidly overfit to fake patterns observed during training, severely undermining generalization to unseen generation methods. A conceptual comparison between previous approaches and our framework is illustrated in Fig. 1.

To address these challenges, we propose a novel detection framework that explicitly models the relational structure between real and synthetic images. Instead of treating samples as independent instances, our approach actively mines real–fake pairs by constructing source-guided mappings or leveraging nearest-neighbor relationships in the CLIP embedding space, thereby uncovering generation-induced discrepancies between semantically aligned samples. To sensitively capture the subtle differences introduced during the generative process, we introduce a pair-wise discrepancy learning objective equipped with a pair-aware regularization mechanism, which stabilizes optimization and mitigates overfitting during discrepancy amplification. In addition, to enhance generalization while preserving the rich representational capacity of pre-trained models, we adopt a bias-only fine-tuning scheme, enabling robust adaptation without disrupting pretrained semantic priors.

In summary, our main contributions are as follows:

- We propose a novel data-centric framework that mines inherent correspondences between forensic and real datasets to construct explicit real-fake data pairs.

- We introduce a pair-wise discrepancy learning with discrepancy inversion mechanism to capture subtle generation-induced artifacts and mitigate overfitting.

- We employ a lightweight fine-tuning strategy that freezes the backbone network and updates only bias parameters to adapt the model to forgery detection.

- Extensive experiments show that our method outperforms existing approaches and generalizes well to cross-generator and cross-domain detection settings.

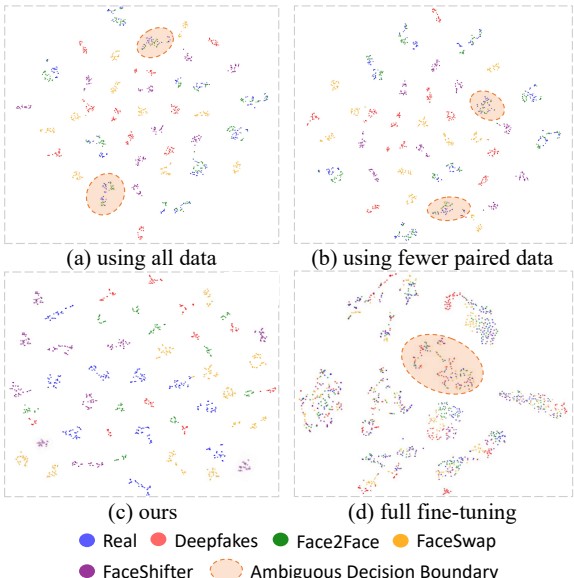

(a) using all data     (b) using fewer paired data

(c) ours     (d) full fine-tuning

● Real   ● Deepfakes   ● Face2Face   ● FaceSwap
● FaceShifter   ⬭ Ambiguous Decision Boundary

*Figure 2.* **Motivating observations. (a)** Unpaired training generalizes poorly. **(b)** Semantic pairing improves generalization with fewer samples. **(c)** Enlarging pair discrepancy enhances separability. **(d)** Full fine-tuning harms generalization.

## 2. Motivation and Analysis

Detecting AI-generated images (AIGI) has become an increasingly important topic. Existing works typically formulate AIGI detection as a standard binary classification problem, treating real and synthetic images as independent samples, and often obtain unsatisfactory generalization performance when faced with unseen fake patterns. Our preliminary investigation suggests that this limitation stems from an overlooked structural property of generative data: synthetic images are not independent instances but are implicitly paired with real images sharing the same semantic source. As a result, naive detectors tend to overfit to superficial instance-level cues rather than capturing the subtle discrepancies within real-fake pairs, thereby undermining their generalization ability to new generative methods. To address these issues, we formulate three key research questions (RQs) below.

**RQ-1: Why is explicitly modeling real-fake pairs more effective than treating real and synthetic images as independent samples?** From Fig. 2 (a) and (b), we observe that a detector trained with substantially fewer semantically paired real-fake samples achieves generalization performance comparable to that trained with a much larger amount of unpaired data. This observation indicates that treating real and synthetic images as independent samples overlooks a critical structural property of generative data, namely the latent semantic correspondence between real-fake pairs. Ignoring such correspondence allows the de-

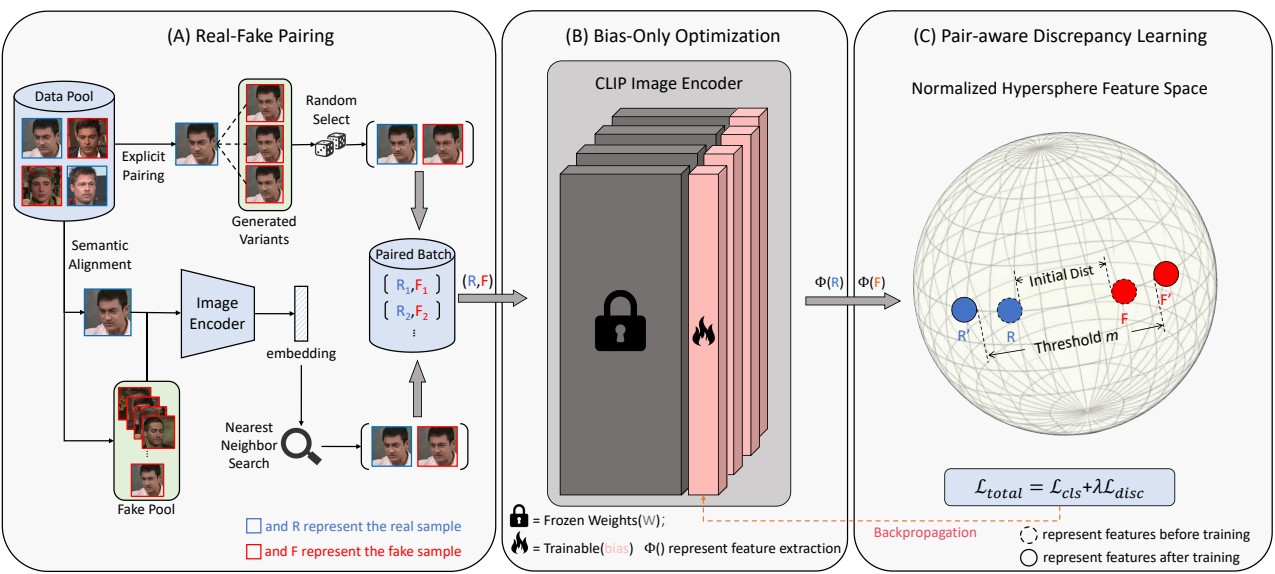

*Figure 3.* **Overview of the proposed BPL framework.** It consists of three stages: **(A)** Real-Fake Pairing, **(B)** Bias-Only Optimization on a frozen CLIP backbone, and **(C)** Pair-aware Discrepancy Learning to enhance real-fake separability.

tector to rely on instance-level semantic shortcuts rather than generation-induced discrepancies, ultimately limiting generalization.

**RQ-2: Why does enlarging the discrepancy within real-fake pairs improve discriminability and generalization?**
Given **RQ-1**, a key insight is that simply using paired data as training samples is insufficient if the relational discrepancies between paired instances remain subtle. Modern generative models often preserve high-level semantics while introducing only weak and entangled artifacts, causing real-fake pairs to remain close in the representation space. Without sufficient discrepancy amplification, standard learning objectives may fail to distinguish such hard pairs, leading to ambiguous decision boundaries and overfitting to superficial instance-level patterns (see Fig. 2 (b)). Therefore, enlarging the discrepancy within real-fake pairs encourages discrimination based on generation-specific cues rather than semantic similarity (see Fig. 2 (c)).

**RQ-3: Why is bias-only adaptation preferable to full or low-rank fine-tuning for preserving generalization in paired detection settings?** A straightforward approach is to fine-tune the backbone with paired supervision. However, full fine-tuning can distort pre-trained semantic representations and degrade generalization, as shown in Fig. 2 (d). Similarly, low-rank fine-tuning still updates projection weights, which risks disrupting the pre-trained knowledge. To preserve the pre-trained semantic manifold, we adopt bias-only fine-tuning that freezes all projection weights and updates only bias terms, enabling effective relational discrimination with minimal disruption to semantic geometry.

Detailed comparisons confirming this advantage over low-rank fine-tuning are provided in Table 7.

The primary goal is to learn a discriminative function $f_\theta$, parameterized by weights $\theta$, that generalizes to samples synthesized by previously unseen architectures. To this end, we propose a data-centric framework that leverages inherent real-fake correspondences to facilitate robust and generalizable forgery detection, as illustrated in Fig. 3.

In the following parts, we detail the design and implementation of each component.

## 3. Methodology

### 3.1. Real-Fake Pairing via Latent Semantic Alignment

Our pair-aware formulation assumes access to real–fake pairs that share the same semantic source. However, in most practical settings, such pairing is not explicitly observable. Instead, the correspondence between real and synthetic images is latent and must be inferred. We therefore treat real–fake pairing not as a fixed preprocessing step, but as a latent semantic alignment problem induced by the generative process.

Let $\mathcal{R} = \{x_i^r\}_{i=1}^{N_r}$ denote a set of real images and $\mathcal{F} = \{x_j^f\}_{j=1}^{N_f}$ denote a set of synthetic images, where $N_r$ and $N_f$ denote the number of instances in $\mathcal{R}$ and $\mathcal{F}$. Each synthetic image $x^f$ is generated from an unobserved real semantic source $x^r$, which we model as a latent variable.

Our goal is to infer a pairing operator

$$\Pi : \mathcal{F} \to \mathcal{R}, \tag{1}$$

such that $\Pi(x^f)$ recovers a real image that best matches the semantic content of $x^f$, up to generation-induced distortions.

Under this view, the quality of a pair is not determined by exact pixel-level correspondence, but by semantic consistency, which is necessary for learning generation-specific discrepancies.

**Explicit Pairing via Observable Generation Traces.** In some datasets, the latent source variable is partially or fully observed. For instance, editing-based generation pipelines or identity-preserving synthesis explicitly record the real image used during generation. In such cases, the pairing operator $\Pi(\cdot)$ is directly instantiated as:

$$\Pi(x_j^f) = x_{s(j)}^r, \tag{2}$$

where $s(j)$ denotes the recorded source index for $x_j^f$.

We emphasize that even in this setting, our method does not rely on additional supervision: the pairing information is a natural byproduct of the generative process rather than a manually curated label. This provides a high-confidence instantiation of the latent alignment assumed by our formulation.

**Semantic Alignment via Cross-distribution Retrieval.** In the absence of explicit generation traces, the source variable must be inferred. We observe that although generation alters low-level statistics, it largely preserves high-level semantics. This motivates us to infer $\Pi(\cdot)$ by semantic alignment across the real and synthetic distributions.

Especially, we use a pretrained CLIP encoder $\phi(\cdot)$ to embed both real and synthetic images into a shared semantic space. All embeddings are $\ell_2$-normalized, such that semantic similarity is measured as angular proximity on the unit hypersphere:

$$\mathbf{z} = \frac{\phi(x)}{\|\phi(x)\|_2}. \tag{3}$$

For each synthetic image $x_j^f$, we define the inferred source as the real image that maximizes semantic similarity by

$$\Pi(x_j^f) = x_{s(j)}^r = \arg\max_{x_i^r \in \mathcal{R}} \ \phi(x_i^r)^\top \phi(x_j^f). \tag{4}$$

where $\phi(x_i^r)^\top \phi(x_j^f)$ measures semantic similarity between real and synthetic images in the shared embedding space. This retrieval is performed across domains, with queries drawn from the synthetic distribution and candidates drawn from the real distribution, rather than within a single dataset.

As a result, each inferred pair $x_j^f, x_{s(j)}^r$ shares similar high-level semantics but is sampled from different distributions, namely $x_j^f \in \mathcal{F}$ and $x_{s(j)}^r \in \mathcal{R}$. The distribution shift between the two is primarily induced by the forgery generator. Consequently, cross-distribution retrieval yields pairs that reflect semantic alignment under generator-induced shift, making them particularly suitable for learning generation-specific discrepancies instead of spurious instance-dependent patterns.

### 3.2. Pair-aware Discrepancy Learning

Having constructed semantically aligned real-fake pairs $(x_j^f, x_{s(j)}^r)$ through latent semantic alignment, we now leverage these pairs to learn generation-specific discrepancies that are otherwise difficult to capture under standard sample-wise supervision.

Traditional forensic training treats real and fake images as independent instances, optimizing a binary classification objective on $\mathcal{R} \cup \mathcal{F}$. However, such formulations often encourage detectors to exploit spurious cues or dataset-specific correlations, rather than focusing on subtle artifacts introduced by the generative process.

In contrast, our structured pair set $\mathcal{P} = \{(x_{s(j)}^r, x_j^f)\}$ enables a discrepancy-centric learning objective: each fake sample is compared against a semantically matched real counterpart, allowing the model to isolate generator-induced deviations under controlled semantic consistency.

**Pair-aware Representation Learning.** Let $f_\theta(\cdot)$ denote the trainable forensic encoder. For each paired sample $(x_{s(j)}^r, x_j^f)$, we obtain normalized feature representations:

$$\mathbf{z}_{s(j)}^r = \frac{\phi(x_{s(j)}^r)}{\|\phi(x_{s(j)}^r)\|_2}, \quad \mathbf{z}_j^f = \frac{\phi(x_j^f)}{\|\phi(x_j^f)\|_2}. \tag{5}$$

Since the paired images share high-level semantics, their feature differences primarily reflect generation-specific artifacts rather than semantic variance.

**Contrastive Discrepancy Objective.** To sensitively capture subtle discrepancies introduced during the generation process, inspired by recent success in contrastive deepfake detection (Liu et al., 2023; Larue et al., 2023; Hong et al., 2024), we adopt a contrastive learning paradigm that directly operates over paired samples.

For a paired batch corresponding to features $\{(\mathbf{z}_{s(j)}^r, \mathbf{z}_j^f)\}_{j=1}^B$, we first define an individual margin-based loss $\mathcal{L}_i$ for each pair $i$:

$$\mathcal{L}_i = -\|\mathbf{z}_i^r - \mathbf{z}_i^f\|_2 \,, \tag{6}$$

where $\mathbf{z}_i^r$ and $\mathbf{z}_i^f$ denote the feature representations of the real and synthetic samples, respectively. This objective

enlarges the distance within each pair primarily reflects generation-induced distortions, making the learned representation more sensitive to forgery.

**Pair-wise Discrepancy Inversion for Mitigating Overfitting.** Although maximizing the pair-wise discrepancy is beneficial for amplifying generation-induced artifacts, we find that continuously minimizing $\mathcal{L}_i$ may lead to overfitting. In particular, once the distance between a paired real-fake sample becomes sufficiently large, further separation tends to encourage the encoder to memorize training-specific forgery cues, which degrades generalization to unseen manipulation patterns.

To mitigate this issue, we propose a simple yet effective **pair-wise discrepancy inversion** strategy. For each paired sample $i$, we measure the pair discrepancy as

$$d_i = \|\mathbf{z}^r_{s(i)} - \mathbf{z}^f_i\|_2. \tag{7}$$

We then define a discrepancy threshold $m$ that characterizes a "sufficient" level of separation. When $d_i$ is below the threshold, the model is encouraged to enlarge the pair discrepancy; however, once $d_i$ exceeds $m$, we invert the loss to prevent excessive separation by

$$\mathcal{L}_i = \begin{cases} -\|\mathbf{z}^r_{s(i)} - \mathbf{z}^f_i\|_2, & d_i < m, \\ \|\mathbf{z}^r_{s(i)} - \mathbf{z}^f_i\|_2, & d_i \geq m. \end{cases} \tag{8}$$

Intuitively, this inversion mechanism allows the model to amplify generation-induced discrepancies in early training, while automatically regularizing the optimization once the discrepancy becomes overly large. As a result, the learned representation avoids over-separating paired samples and focuses on robust discrepancy cues that generalize beyond the seen generators.

In addition to the discrepancy-driven objective in Eq. 8, we introduce a supervised binary classification loss to enforce reliable real-fake discrimination. Specifically, given the semantically aligned real-fake feature pair $\mathbf{z}^r_{s(i)}$ and $\mathbf{z}^f_i$, we apply a lightweight binary classifier $g(\cdot)$ to obtain their predicted fake probabilities:

$$p^r_{s(i)} = g(\mathbf{z}^r_{s(i)}), \qquad p^f_i = g(\mathbf{z}^f_i), \tag{9}$$

where $p \in [0, 1]$ denotes the probability of being fake.

Since real samples are labeled as 0 and fake samples as 1, we define the binary cross-entropy supervision loss as

$$\mathcal{L}_{cls} = -\log(1 - p^r_{s(i)}) - \log p^f_i, \tag{10}$$

which encourages $\mathbf{z}^r_{s(i)}$ to be classified as real and $\mathbf{z}^f_i$ as fake.

The final training objective is a weighted combination of the discrepancy-driven loss and the classification loss:

$$\mathcal{L}_{total} = \lambda \mathcal{L}_{disc} + \mathcal{L}_{cls}, \tag{11}$$

where $\mathcal{L}_{disc} = \frac{1}{B} \sum_i \mathcal{L}_i$ denotes the discrepancy loss in Eq. 8, and $\lambda$ controls the relative strength of discrepancy-driven objective.

### 3.3. Bias-only Optimization

Directly fine-tuning all parameters of the encoder based on Eq. 11 can be suboptimal. Because the objective explicitly encourages increased separation between semantically aligned real-fake pairs, full fine-tuning provides excessive model capacity to over-amplify such discrepancies. As a result, the encoder may adapt to generator-specific artifacts or dataset-dependent cues, which undermines robustness to unseen fake patterns.

Recent popular approaches such as Effort (Yan et al., 2025b) seek to improve generalization by decomposing pretrained weights into orthogonal semantic and residual subspaces via singular value decomposition (SVD), and adapting only the residual components for forgery modeling. While such designs constrain the trainable space, they still introduce explicit learnable directions dedicated to forgery patterns.

From a functional perspective, the parameters of a neural layer can be conceptually divided into those that define the principal representational geometry and bias terms that control activation shifting. SVD-based methods focus on decomposing and adapting the former, explicitly modifying the underlying representational subspace to accommodate forgery patterns. However, explicitly adapting the representational subspace still modifies the geometric structure of pretrained features and introduces additional learnable directions dedicated to modeling forgery characteristics, resulting in disrupting the pretrained semantic structure that enables transferability across diverse forgery distributions.

To avoid these issues, we adopt a more minimal adaptation strategy by freezing the pretrained weights and optimizing only the bias terms. Bias-only tuning enables lightweight activation re-centering to account for generation-induced distributional shifts, without introducing new representational directions or disrupting semantic geometry. This design explicitly decouples general-purpose representation learning from forgery-specific adaptation, leading to improved robustness under our discrepancy-driven training regime.

Formally, let $f_\theta(\cdot)$ denote the pretrained encoder, whose parameters can be partitioned into weight parameters $\theta_W$ and bias parameters $\theta_b$. Under the bias-only optimization scheme, we freeze $\theta_W$ and update only the bias terms, i.e.,

$$\theta_b \leftarrow \theta_b^{(0)} - \eta \nabla_{\theta_b} \mathcal{L}_{total}, \tag{12}$$

where $\theta^{(0)}$ denotes the pretrained initialization, $\eta$ is the learning rate, $\nabla_{\theta_b}$ denotes the gradient of the loss with respect to the bias parameters, and $\mathcal{L}_{total}$ is the overall training objective defined in Eq. 11.

By restricting optimization to bias parameters, the representational geometry induced by the pretrained weights is preserved throughout training. The bias updates act as additive shifts on intermediate activations, providing a low-capacity yet effective mechanism to adapt to generation-induced distributional discrepancies. Compared to residual subspace adaptation, bias-only tuning introduces significantly fewer trainable parameters and avoids learning new forgery-specific directions in the feature space, resulting in superior generalization across diverse unseen fake patterns.

# 4. Experiments

## 4.1. Deepfake Image Detection

**Implementation Details.** We adopt a bias-only adaptation scheme on CLIP ViT-L/14 (Radford et al., 2021), in which only the bias parameters are updated while all backbone weights remain frozen. The data pre-processing and training procedure is aligned with the DeepfakeBench protocol (Yan et al., 2023b). Optimization is performed using Adam (Kingma & Ba, 2015) with a fixed learning rate of 1e-4. We use a batch size of 8 during training and 32 at inference time. Following common practice in deepfake detection (Cheng et al., 2024; Shiohara & Yamasaki, 2022; Yan et al., 2024a), we apply standard data augmentations, including Gaussian blur and image compression.

For evaluation, we report video-level Area Under the Curve (AUC), which is the standard metric used in prior deepfake detection studies (Lin et al., 2024; Xu et al., 2023; Shiohara & Yamasaki, 2022). A single prediction score for each video is obtained by averaging the model's frame-level confidence outputs.

**Evaluation Protocols and Datasets.** We evaluate all methods under a cross-manipulation setting within the Celeb-DF (Li et al., 2020) data domain, where models are trained and tested on forgery techniques generated from a shared data distribution. Specifically, we adopt the recently released DF40 benchmark (Yan et al., 2024b), which consists of diverse manipulation methods constructed exclusively within the Celeb-DF environment. This design allows the evaluation to isolate the effect of manipulation variation while avoiding domain-shift-induced bias.

To facilitate a comprehensive comparison, we benchmark our approach against **9 competitive detectors**, covering both widely adopted classical methods, such as F3Net (Wang et al., 2023), SPSL (Liu et al., 2021), SRM (Luo et al., 2021), CORE (Ni et al., 2022), RECCE (Cao et al.,

2022), and SBI (Shiohara & Yamasaki, 2022), as well as more recent high-performing approaches introduced after 2023, including UCF (Yan et al., 2023a), IID (Huang et al., 2023), TALL (Xu et al., 2023), and Effort (Yan et al., 2025b). All models are trained using Celeb-DF (c23) and evaluated on forgery data produced by unseen manipulation methods. Quantitative results are reported in Tab. 1.

As shown in Tab. 1, our method consistently outperforms existing approaches across most cross-manipulation evaluation settings. BPL achieves the highest average AUC of 88.86%, surpassing the strongest baseline Effort by 2.57%, and demonstrating clear advantages over traditional artifact-based detectors.

When comparing different detection paradigms, conventional low-level artifact-based methods exhibit limited robustness under cross-manipulation settings, with performance varying substantially across manipulation types. Although some methods perform well on specific manipulations, their overall generalization remains weak, reflecting a strong reliance on manipulation-specific artifacts. In contrast, VLM-based approaches achieve markedly stronger overall performance, highlighting the benefit of high-level semantic representations. Noticeable performance variations across manipulations still persist for Effort, indicating residual sensitivity to dataset or manipulation bias.

Across different manipulation methods, BPL maintains consistently strong performance and achieves the best results on the majority of evaluated categories. In particular, the performance of our method remains stable even on challenging manipulations where other approaches suffer from clear degradation, suggesting improved robustness to variations in manipulation mechanisms. These gains can be attributed to the proposed paired bias-only adaptation strategy. By explicitly leveraging aligned real–fake pairs, our method suppresses spurious manipulation-specific cues while emphasizing semantically meaningful discrepancies that violate human perception. As a result, BPL achieves improved cross-manipulation generalization without relying on dataset-specific optimization.

## 4.2. Hyperparameter Sensitivity Analysis

Fig. 4 shows the sensitivity of AP performance to different hyperparameter settings. Overall, the proposed method exhibits stable performance, with AP remaining above 90% in most cases, indicating low sensitivity to hyperparameter variations.

**Impact of $\lambda$.** Across different values of $\lambda$, the model consistently achieves strong performance. In particular, $\lambda = 0.4$ yields the highest AP in most settings, with a peak performance of 95.43%, while smaller or larger values lead to slightly lower AP under certain configurations.

*Table 1.* Benchmarking Results of Cross-manipulation Evaluations in terms of AUC Performance (%). The best results are shown in **bold**, and the second-best results are underlined.

| METHOD | SIMSWAP | BLEFACE | UNIFACE | E4S | FACEDAN | FSGAN | INSWAP | MOBSWAP | DFLAB | AVG. |
|---|---|---|---|---|---|---|---|---|---|---|
| F3NET | 46.32 | 65.58 | 48.57 | 29.81 | 65.31 | 74.44 | 66.02 | 64.42 | 74.82 | 59.48 |
| SPSL | 35.22 | 52.97 | 45.19 | 29.94 | 46.83 | 65.64 | 51.67 | 47.96 | 60.55 | 48.44 |
| SRM | 44.80 | 64.83 | 48.73 | 31.46 | 60.51 | 71.54 | 59.35 | 63.63 | 70.51 | 57.26 |
| CORE | 43.34 | 67.67 | 54.48 | 26.24 | 57.65 | 78.88 | 62.88 | 67.59 | 76.49 | 59.47 |
| RECCE | 46.43 | 69.11 | 51.03 | 23.68 | 54.88 | 77.74 | 65.37 | 68.50 | 71.79 | 58.73 |
| SBI | 49.56 | 71.99 | 58.66 | 31.67 | 54.75 | 76.60 | 61.60 | 65.96 | 69.63 | 60.05 |
| UCF | 51.41 | 70.48 | 63.74 | 41.72 | 66.34 | 75.25 | 64.14 | 72.45 | 73.85 | 64.38 |
| TALL | 46.97 | 55.71 | 54.18 | 47.04 | 55.21 | 53.66 | 49.68 | 56.53 | 57.12 | 52.90 |
| EFFORT | 82.10 | 86.91 | 83.32 | **96.02** | 88.84 | 88.80 | 74.27 | 87.03 | 89.35 | 86.29 |
| **BPL(OURS)** | **83.81** | **91.05** | **84.88** | 93.83 | **90.84** | **92.25** | **74.92** | **92.78** | **95.39** | **88.86** |

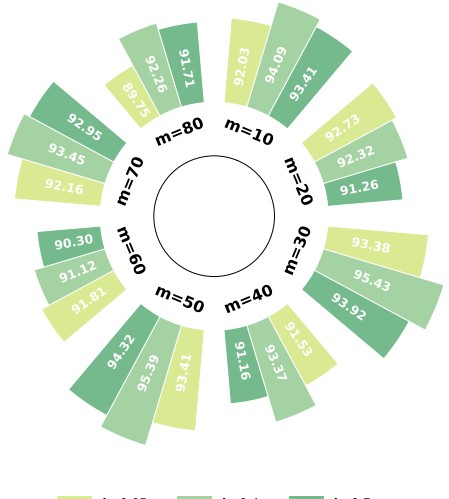

*Figure 4.* **Hyperparameter Sensitivity Analysis.** AP performance (%) under different loss weights $\lambda \in \{0.05, 0.4, 0.7\}$ and margin values $m$ ranging from 10 to 80.

**Impact of margin $m$.** With respect to $m$, intermediate values generally result in higher AP across different $\lambda$ settings, whereas overly large margins cause mild performance degradation. Nevertheless, the overall variation remains limited, suggesting robust behavior with respect to margin selection.

### 4.3. Ablation Analysis

**How effective is bias-only tuning as a foundational strategy?** As shown in Tab. 2, the baseline model that updates all backbone parameters achieves an average AUC of 43.16% across the evaluated datasets. When switching to the bias-only tuning strategy, where the backbone is frozen and only bias parameters are updated, the average AUC increases substantially to 76.22%, yielding an absolute improvement of **+33.06%**. This significant gain indicates that fully fine-tuning large-scale vision foundation models on limited deepfake data can severely impair generalization, whereas bias-only tuning effectively constrains the adaptation space and preserves the transferable representations learned during pre-training.

**What is the impact of incorporating paired data sampling?** Building upon the bias-only configuration, introducing the paired data sampling strategy further improves the average AUC from 76.22% to 81.81%, corresponding to an additional gain of **+5.59%**. This improvement is consistently observed across all evaluated datasets. Even without an explicit contrastive objective, paired sampling serves as an implicit hard example mining mechanism, encouraging the model to focus on subtle forensic discrepancies between semantically aligned real–fake pairs rather than relying on spurious background cues.

**Structurally aligned, high-quality data is preferable to indiscriminate use of all available samples.** In our setting, paired data corresponds to structurally aligned, high-quality training samples, whose advantage lies in emphasizing data quality and structural alignment rather than sheer data quantity. In contrast to the "All Data" setting, which utilizes all available fake frames in an unpaired manner, the paired strategy relies on a carefully selected subset of strictly aligned real–fake pairs. As shown in Tab. 3, applying paired sampling to our method improves the average AUC from 80.20% to 81.12%, despite using fewer training samples. In comparison, this trend does not generalize to Effort, where replacing unpaired data with paired samples leads to a slight performance degradation (from 79.63% to 78.64%), suggesting that merely constraining data volume without an appropriate adaptation mechanism is insufficient. These results indicate that, when coupled with our bias-only adaptation, paired samples serve as high-quality exemplars that highlight discriminative forensic cues while mitigating redundant or noisy supervision introduced by large-scale unpaired data.

**Is it necessary to explicitly push apart paired samples?** Finally, integrating the contrastive loss completes the proposed framework and achieves the best overall performance. With the contrastive constraint enabled, the average AUC increases from 81.81% to 88.86%, yielding an additional

*Table 2.* **Ablation study of our framework components,** including bias-only fine-tuning, paired data sampling, and the pairwise objective $\mathcal{L}_{\text{disc}}$. All models are trained on Celeb-DF (c23) and evaluated using AUC (%).

| Bias-only | Paired | $\mathcal{L}_{\text{pair}}$ | SimSwap | BleFace | UniFace | e4s | FaceDan | FSGAN | InSwap | MobSwap | DFLab | Avg. |
|---|---|---|---|---|---|---|---|---|---|---|---|---|
| × | × | × | 43.43 | 45.24 | 53.63 | 34.87 | 38.41 | 40.07 | 40.84 | 40.60 | 51.37 | 43.16 |
| √ | × | × | 71.40 | 75.01 | 74.22 | 81.11 | 78.76 | 83.10 | 62.37 | 75.19 | 84.95 | 76.22 |
| √ | √ | × | 78.94 | 83.92 | 80.42 | 88.41 | 80.43 | 85.14 | 69.11 | 82.63 | 87.31 | 81.81 |
| √ | √ | √ | 83.81 | 91.05 | 84.88 | 93.83 | 90.84 | 92.25 | 74.92 | 92.78 | 95.39 | 88.86 |

*Table 3.* **Impact of Paired Training Strategy.** We compare paired training with the "All Data" setting on Effort and **Ours**. Results are reported in AUC (%). "All Data" denotes training with all available fake frames in an unpaired manner, while "Paired" uses strictly aligned real–fake pairs.

| Method | Data Strategy | SimSwap | e4s | InSwap | DFLab | Avg. |
|---|---|---|---|---|---|---|
| Effort | All Data | 83.99 | 79.57 | 72.57 | 82.37 | 79.63 |
| | Paired | 82.87 | 80.31 | 72.71 | 78.67 | 78.64 |
| **Ours** | All Data | 81.30 | 83.26 | 70.75 | 85.48 | 80.20 |
| | Paired | 81.83 | 81.40 | 71.67 | 89.59 | 81.12 |

improvement of **+7.05%**. This result demonstrates that paired data alone is insufficient to fully leverage the underlying correspondence; explicit geometric constraints are necessary to ensure meaningful separation between real and fake representations in the embedding space. Unlike standard classification losses that primarily focus on inter-class separability, the proposed contrastive objective directly maximizes the margin between paired samples, leading to more robust and discriminative forensic features.

### 4.4. Generalization Analysis

In this section, we conduct a comprehensive evaluation to verify the robustness of our method across two critical dimensions: stability under extreme data scarcity and generalization capability across unseen datasets.

**Generalization to Extreme Data Scarcity.** We evaluate robustness in label-limited regimes by comparing models trained under the same protocol using either 10% or 100% of the available training data. As shown in Tab. 4, Effort suffers a noticeable degradation when training data is reduced, with its average AP dropping by **4.26%**. In contrast, our method exhibits substantially higher stability, showing only a marginal decline of **1.35%** when trained with 10% of the data. Notably, even under this extreme data scarcity, our approach consistently outperforms Effort trained on the full dataset across all evaluated benchmarks. These results demonstrate that the proposed bias-only adaptation effectively constrains the optimization space, enabling reliable generalization in low-resource settings.

**Generalization to Cross-Datasets.** Beyond cross-manipulation robustness, we further evaluate generalization to completely unseen datasets. All models are trained on Celeb-DF and directly tested on three external benchmarks:

*Table 4.* **Robustness under Data Scarcity.** Comparison of AP Score (%) between Effort and Ours when training data is reduced from 100% to 10%.

| Method | Ratio | DFDC | DFDCP | SimSwap | FSGAN | **Avg.** | Gap |
|---|---|---|---|---|---|---|---|
| Effort | 100% | 70.67 | 80.82 | 93.65 | 96.44 | 85.39 | ↓4.26 |
| | 10% | 62.86 | 76.64 | 90.67 | 94.35 | 81.13 | |
| **Ours** | 100% | 82.84 | 86.55 | 94.45 | 97.52 | 90.34 | ↓**1.35** |
| | 10% | 79.12 | 84.62 | 94.56 | 97.65 | 88.99 | |

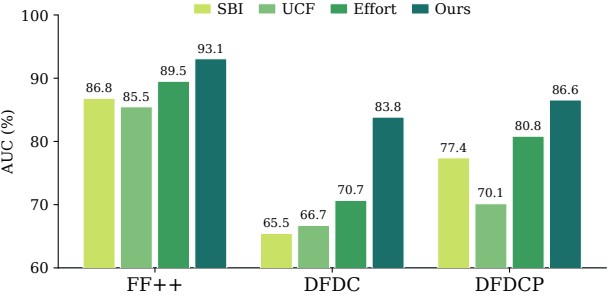

*Figure 5.* Cross-dataset generalization results when trained on Celeb-DF and evaluated on unseen datasets.
FaceForensics++ (FF++) (Rössler et al., 2019), DFDC (detection challenge., 2020), and DFDCP (Dolhansky et al., 2019). As illustrated in Fig. 5, our method consistently outperforms strong generalization baselines across all target domains. On FF++, our approach achieves an AUC of 93.1%, surpassing Effort by **+3.6%**, and exceeding SBI and UCF by **+6.3%** and **+7.6%**, respectively. The performance gap becomes more pronounced on DFDC, where our method attains 83.8%, outperforming Effort by **+13.1%**, SBI by **+18.3%**, and UCF by **+17.1%**. Similarly, on DFDCP, our method reaches 86.6%, yielding improvements of **+5.8%**, **+9.2%**, and **+16.5%** over Effort, SBI, and UCF, respectively. These consistent gains across diverse and unseen datasets indicate that the proposed manifold-preserving bias tuning strategy captures intrinsic forgery-related patterns, rather than overfitting to the source distribution or dataset-specific artifacts.

## 5. Conclusion

In this paper, we present a minimalist framework for generalizable face forgery detection via Bias-only Adaptation. Unlike complex architectural disentanglement, our proposed Paired Data Mining achieves data-driven decoupling by leveraging hard-mined pairs to isolate forensic artifacts

while strictly preserving the pre-trained semantic manifold. Extensive experiments confirm that this paradigm achieves superior cross-dataset generalization and robustness in data-scarce regimes, significantly outperforming existing parameter-efficient baselines.

## Impact Statement

This paper presents work whose goal is to advance the field of machine learning, with a focus on improving the robustness of AI-generated image detection. The potential societal implications of this research are consistent with those commonly associated with advances in detection and robustness methods, and we do not identify any specific ethical concerns that require additional discussion.

## Acknowledgements

This work was supported by the Excellent Young Scientists Fund Program (Overseas) of the Shandong Provincial Natural Science Foundation (2026HWYQ-007) and the Youth Innovation Science and Technology Support Program for Higher Education Institutions of Shandong Province (2025KJH105).

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

# A. Related Work

Existing methodologies for AI-generated image detection can be primarily categorized into two distinct paradigms based on the level of abstraction of the discriminative cues they exploit. The first category predominantly focuses on leveraging low-level, often imperceptible visual artifacts that are residual from the generative synthesis process. The second category shifts towards utilizing high-level semantic priors inherent in large-scale foundation models to identify subtle inconsistencies.

## A.1. Low-Level Visual Artifact-Based Methods

Traditional artifact-centric detection methods typically operate on the premise that real and synthetic images possess inherent differences in low-level visual statistics. To capture these subtle cues, recent studies have introduced specialized architectures targeting specific local or structural generative traces. For instance, FatFormer (Liu et al., 2024) integrates forgery-aware adapters into Transformer (Vaswani et al., 2017) blocks to attend to high-frequency anomalies; SAFE (Li et al., 2025) employs image transformations to uncover local texture irregularities often masked by standard down-sampling; NPR (Tan et al., 2024) introduces the concept of Neighboring Pixel Relationships to capture generalized structural artifacts stemming from up-sampling operations in generative networks. The dominant paradigm for deploying these mechanisms involves leveraging deep CNN or Transformer backbones pre-trained on large-scale datasets (e.g., EfficientNet (Tan & Le, 2019), Xception (Chollet, 2017)) and further fine-tuning them for forgery detection.

Beyond directly exploiting artifacts from individual images, recent studies have explored constructing paired real-fake data to alleviate dataset bias, typically by enforcing alignment through reconstruction under specific generation pipelines. AlignedForensics (Rajan et al., 2025) and DDA (Chen et al., 2025) are representative works along this line, where paired samples are generated via model-specific reconstruction processes to enable fine-grained discrepancy learning. Similarly, recent work has also utilized paired training data to mitigate shortcut learning and improve generalization (Yermakov et al., 2026). These methods encourage detectors to focus on low-level discrepancies introduced by particular generative or reconstruction pipelines, reducing spurious dataset correlations. Despite their effectiveness in debiasing, they primarily constrain detectors to learn generator-specific low-level artifacts, limiting generalization beyond the reconstruction models used.

While effective on specific benchmarks, low-level artifact-based methods in general face significant limitations. In attempting to fit dataset- or generator-specific artifact patterns, models often suffer from catastrophic forgetting, degrading the generalized visual representations acquired during pre-training and resulting in poor cross-dataset generalization. Moreover, these methods frequently struggle against advanced generation architectures that intentionally suppress such traces, motivating the exploration of more abstract, semantic-level cues.

## A.2. Vision-Language Model Based Approaches

Motivated by these limitations, recent research has shifted towards leveraging high-level semantic priors inherent in vision foundation models (VFMs), particularly vision-language models such as CLIP (Radford et al., 2021). The core premise is that photorealistically synthesized images, although visually convincing, often contain subtle semantic or logical inconsistencies that violate human perception or commonsense knowledge learned by large-scale models during pre-training.

Methods within this paradigm exploit semantic priors in different ways. AIDE (Yan et al., 2025a) utilizes CLIP as an expert model through a multi-expert framework; Effort (Yan et al., 2025b) disentangles semantic content and generation artifacts via orthogonal decomposition in the feature space; ConV (Zhang et al., 2025) models natural image distributions in frozen, pre-trained feature spaces; PPL (Yang et al., 2025) further emphasizes patch-level analysis to capture subtle local inconsistencies masked by global semantics. Despite improved generalization, these approaches typically treat samples independently and rely on semantic consistency or distributional modeling, without explicitly exploiting the relational structure between real and synthetic images.

# B. Additional Cross-Manipulation Evaluation Results

As reported in Tab. 5, our method achieves consistently superior performance across most cross-manipulation evaluation settings under the ACC metric. BPL obtains the highest average accuracy of 84.23%, outperforming the strongest baseline Effort by 2.00%, which further validates the effectiveness of our framework beyond AUC.

Similar to the observations under AUC, conventional artifact-based detectors exhibit limited robustness in cross-manipulation

*Table 5.* Benchmarking Results of Cross-manipulation Evaluations in terms of Accuracy (ACC) Performance. The best results are shown in **bold**, and the second-best results are underlined.

| METHOD | SIMSWAP | BLEFACE | UNIFACE | E4S | FACEDAN | FSGAN | INSWAP | MOBSWAP | DFLAB | AVG. |
|---|---|---|---|---|---|---|---|---|---|---|
| F3NET | 33.07 | 58.03 | 37.23 | 28.95 | 57.79 | 69.59 | 59.54 | 55.63 | 57.37 | 50.80 |
| SPSL | 24.42 | 35.17 | 28.94 | 36.50 | 29.91 | 45.78 | 40.25 | 31.89 | 38.00 | 34.54 |
| SRM | 31.23 | 42.89 | 29.87 | 33.10 | 38.23 | 51.93 | 44.94 | 39.51 | 37.74 | 38.83 |
| CORE | 29.64 | 52.49 | 38.01 | 31.28 | 41.73 | 67.93 | 52.08 | 47.66 | 47.16 | 45.31 |
| RECCE | 26.14 | 43.43 | 26.95 | 33.77 | 28.81 | 55.53 | 45.12 | 40.15 | 38.30 | 37.58 |
| SBI | 30.72 | 54.69 | 38.88 | 33.14 | 34.96 | 63.60 | 47.77 | 46.49 | 46.54 | 44.09 |
| UCF | 44.26 | 63.69 | 57.56 | 42.14 | 60.90 | 71.20 | 57.96 | 66.21 | 55.64 | 57.73 |
| TALL | 33.98 | 42.56 | 38.42 | 42.90 | 41.31 | 38.97 | 40.71 | 42.60 | 39.62 | 40.01 |
| EFFORT | 82.87 | 85.89 | **83.55** | 80.31 | 86.53 | 84.76 | 72.71 | 84.78 | 78.67 | 82.23 |
| **OURS** | **83.73** | **86.86** | 83.01 | **80.49** | **86.59** | **85.27** | **74.18** | **87.53** | **90.43** | **84.23** |

scenarios, with accuracy varying substantially across different manipulation types. Although some methods perform competitively on specific generators, their overall generalization remains weak, reflecting a strong dependence on manipulation-specific low-level cues. In contrast, VLM-based approaches demonstrate markedly improved overall accuracy, highlighting the benefit of transferable semantic representations for forgery detection.

Notably, BPL maintains stable performance across diverse manipulations and achieves the best results on the majority of evaluated categories, particularly on challenging cases such as DFLab, where other methods suffer from clear degradation. These gains can be attributed to our paired bias-only adaptation strategy, which suppresses spurious generator-specific correlations while emphasizing semantically aligned discrepancies between real and fake samples, leading to improved cross-manipulation generalization.

*Table 6.* Benchmarking Results of Cross-manipulation Evaluations in terms of AP Performance on the DeepfakeBench Dataset. The best results are shown in **bold**, and the second-best results are underlined.

| METHOD | SIMSWAP | BLEFACE | UNIFACE | E4S | FACEDAN | FSGAN | INSWAP | MOBSWAP | DFLAB | AVG. |
|---|---|---|---|---|---|---|---|---|---|---|
| F3NET | 75.09 | 85.81 | 75.45 | 48.48 | 86.18 | 90.73 | 82.25 | 85.96 | 86.33 | 79.59 |
| SPSL | 72.51 | 82.54 | 77.52 | 50.17 | 78.84 | 88.17 | 75.74 | 80.23 | 75.16 | 75.65 |
| SRM | 77.46 | 86.22 | 77.16 | 49.16 | 83.96 | 88.76 | 78.26 | 84.97 | 81.41 | 78.60 |
| CORE | 75.05 | 86.75 | 79.71 | 47.35 | 82.35 | 92.00 | 79.45 | 85.75 | 85.49 | 79.32 |
| RECCE | 75.90 | 87.53 | 77.13 | 46.37 | 79.40 | 91.58 | 80.73 | 86.66 | 83.72 | 78.78 |
| SBI | 78.20 | 89.52 | 82.49 | 49.70 | 80.72 | 91.46 | 79.25 | 86.74 | 83.73 | 80.20 |
| UCF | 79.47 | 88.43 | 83.99 | 57.55 | 86.44 | 90.35 | 80.68 | 89.07 | 83.55 | 82.17 |
| TALL | 78.45 | 82.83 | 81.21 | 61.83 | 82.97 | 81.05 | 72.68 | 83.35 | 70.20 | 77.17 |
| EFFORT | 93.65 | 95.45 | 93.95 | **97.54** | **96.29** | 96.44 | 86.08 | 95.74 | 94.30 | 94.38 |
| **OURS** | **94.94** | **96.87** | **94.81** | 93.16 | 95.36 | **97.30** | **88.40** | **97.37** | **97.85** | **95.12** |

Tab. 6 presents cross-manipulation benchmarking results in terms of AP. Our method consistently achieves the strongest overall performance, reaching the highest average AP of 95.12%, surpassing the best competing baseline Effort by 0.74%. This demonstrates that BPL not only improves ranking quality but also provides more reliable precision-recall behavior under unseen manipulation settings.

Across different detection paradigms, low-level artifact-based methods show weaker robustness when evaluated on manipulation types not observed during training, indicating their reliance on generator-specific artifacts. Meanwhile, VLM-based detectors benefit from stronger and more transferable semantic representations, yielding substantially improved AP across most manipulation categories.

Moreover, BPL exhibits consistently strong performance across all evaluated forgery types and achieves the best results on the majority of manipulation methods, especially on challenging generators such as MobSwap and DFLab. Such improvements stem from the proposed paired discrepancy learning formulation together with bias-only adaptation, which explicitly leverages aligned real-fake pairs to enforce semantically meaningful separability while reducing sensitivity to manipulation-specific spurious cues.

## C. Comparison with Standard PEFT Methods

To further validate the effectiveness of our bias-only optimization strategy, we compare BPL with standard Parameter-Efficient Fine-Tuning (PEFT) methods, including LoRA with varying rank sizes ($r \in \{1, 4, 8\}$) and LayerNorm Tuning (LN_Tuning).

As shown in Table 7, while standard PEFT methods achieve reasonable performance, they generally struggle to match the generalization capabilities of our approach. BPL consistently outperforms all evaluated PEFT baselines across various unseen manipulations, achieving the highest average AUC of 88.86%. This confirms that our lightweight bias-only tuning is more effective at resolving distributional shifts while strictly preserving pre-trained semantic priors compared to conventional adapter-based or low-rank subspace modifications.

*Table 7.* Comparison with Standard PEFT Methods in terms of AUC Performance (%). The best results are shown in **bold**, and the second-best results are underlined.

| METHOD | SIMSWAP | BLEFACE | UNIFACE | E4S | FACEDAN | FSGAN | INSWAP | MOBSWAP | DFLAB | AVG. |
|---|---|---|---|---|---|---|---|---|---|---|
| LORA ($r = 1$) | 54.14 | 54.45 | 54.26 | 54.33 | 55.10 | 54.31 | 53.72 | 54.37 | 55.20 | 54.43 |
| LORA ($r = 4$) | 76.64 | 79.06 | 78.21 | 72.88 | 76.93 | 80.24 | 68.49 | 81.37 | 61.37 | 75.02 |
| LORA ($r = 8$) | 71.42 | 75.09 | 74.42 | 68.66 | 74.15 | 74.19 | 67.62 | 78.32 | 67.48 | 72.37 |
| LN_TUNING | 72.00 | 77.67 | 71.45 | 88.25 | 76.33 | 80.60 | 63.74 | 77.06 | 85.03 | 76.90 |
| **BPL(OURS)** | **83.81** | **91.05** | **84.88** | **93.83** | **90.84** | **92.25** | **74.92** | **92.78** | **95.39** | **88.86** |

