# OpenReview forum: "BPL: Generalizable Deepfake Detection via Bias-only Pair-aware Learning"
_ICML.cc/2026/Conference — ICML 2026 regular_

### Official Review · Reviewer_vdsM · 2026-03-02

**Soundness:** 3
**Presentation:** 3
**Significance:** 3
**Originality:** 3
**Overall Recommendation:** 4
**Confidence:** 4

**Summary:**

This paper proposes a new perspective for training deepfake detectors. The authors claim that the real-fake images are paired w.r.t. semantic content, and propose a (1) pair-wise discrepancy learning strategy and (2) bias-only fine-tuning method to train the detector. Experiments on DF40 show the effectiveness of the proposed method.

**Compliance With Llm Reviewing Policy:**

Affirmed.

**Final Justification:**

My major concerns (more rigorous evaluation, comparisons with recent methods) are addressed, and I’m willing to raise my score. For other concerns, mostly are not directly provided in the rebuttal (due to format constraints). Based on this, I respect other reviewers' and AC's final judgment.

**Key Questions For Authors:**

My major concerns are listed in Weakness 1-3. I will raise my score if these concerns are addressed.

Some other questions:
1. What are the results of using CLIP for NN retrieval? Since CLIP is trained for “high-level semantics” (e.g., face, apple, chair), how would it perform when retrieving different faces? Could the author provide some examples of the paired/unpaired results? What if using a different pretrained model?
2. In Fig. 2 (c), the real samples (blue dots) is clearly much fewer than other subplots. Did you keep the samples consistent when visualizing?
3. If z^r/z^f denotes the features, then what is h^r and h^f in Eq. (6). Are Eq. (6) and (8) using the same variable?

**Limitations:**

no.

The limitations of the method should be discussed.

**Strengths And Weaknesses:**

**Strength**
1. The paper is well presented and easy to follow.
2. The main idea (paired learning) is reasonable.
3. Rather than adding incremental modules, the proposed method (discrepancy learning and bias-only fine-tuning) is interesting and fundamental.

**Weakness**

My main concerns are the insufficient experiments:
1. In Table 2, the training/testing real images are from the same domain (i.e., CDF), hindering the understanding of the method’s effectiveness. Since the method is supposed to be a universal training strategy, more rigorous evaluation should be conducted, e.g., (1) how about using DF40(FF++) to train and DF40(CDF) to test, and (2) protocol introduced by a recent dataset[1].
2. The compared methods in the main experiment are insufficient. Some recent methods that also share similar ideas should be considered, e.g., DDA[2] and AlignedForensics[3].
3. What if the training data doesn’t contain clear paired examples (e.g., no matching IDs, or large differences in style/resolution between real and fake images)?

Some other concerns:

4. The evaluation datasets are mainly FS and FR, which do not include Entire Face Synthesis, more evaluation should be conducted.
5. Why is AUC used in the main experiment and AP used in the ablation studies?
6. Could the author add the LoRA fine-tuned baseline? That is, using LoRA for backbone fine-tuning and L_cls for supervision.


[1] Veritas: Generalizable deepfake detection via pattern-aware reasoning

[2] Dual Data Alignment Makes AI-Generated Image Detector Easier Generalizable

[3] Aligned Datasets Improve Detection of Latent Diffusion-Generated Images

---

> ### Author Rebuttal · Authors · 2026-03-31
>
> **Q1. Regarding Rigorous Cross-domain Evaluation and Universal Training Strategy**
>
> >We fully agree that testing on unseen domains is essential to validate BPL as a universal training strategy.
>
> **Cross-Domain Evaluation (Training on CDF, Testing on FF++):**
> We evaluated our model's cross-domain performance by training exclusively on CDF and testing on the unseen FF++ dataset.
>
> >Below are the AUC (%) results of our model in both in-domain and cross-domain evaluations, demonstrating only a slight performance drop in the cross-domain setting.
>
> |  | SimSwap | BleFace | UniFace | e4s | FaceDan | FSGAN | InSwap | MobSwap | DFLab | **Avg.** |
> | :--- | :---: | :---: | :---: | :---: | :---: | :---: | :---: | :---: | :---: | :---: |
> | **In-domain** | 83.81 | 91.05 | 84.88 | 93.83 | 90.84 | 92.25 | 74.92 | 92.78 | 95.39 | 88.86 |
> | **Cross-domain** | 81.86 | 76.33 | 85.51 | 90.12 | 78.63 | 84.75  | 72.84 | 84.09 | 94.28 | 85.62 |
>
> **Regarding the Evaluation Protocol from VERITAS:**
>
> >Due to time constraints, VERITAS evaluation is deferred to the revision. To address cross-domain concerns, we report zero-shot results (AUC %) on unseen DFDC and DFDCP. Trained exclusively on CDF, BPL outperforms Effort:
>
> |  | DFDC | DFDCP |
> | :--- | :---: | :---: |
> | Effort | 68.52 | 76.52 |
> | BPL (Ours) | 78.08 | 82.59 |
>
> **Q2. Comparison with recent methods like DDA and AlignedForensics**
>
> >To address your concern, we evaluate DDA and AlignedForensics using the same training dataset as BPL, which is composed of real images and their corresponding fake samples synthesized via DDA. The AUC (%) results across 9 manipulation methods are summarized below:
>
> |  | SimSwap | BleFace | UniFace | e4s | FaceDan | FSGAN | InSwap | MobSwap | DFLab | **Avg.** |
> | :--- | :---: | :---: | :---: | :---: | :---: | :---: | :---: | :---: | :---: | :---: |
> | DDA | 69.09 | 66.68 | 87.01 | 56.38 | 40.42 | 54.99 | 73.50 | 32.28 | 32.07 | 56.94 |
> | AlignedForensics | 65.89 | **80.54** | **90.21** | 55.01 | 73.48 | 56.57 | **77.45** | 45.33 | 60.84 | 67.28 |
> | BPL (Ours) | **77.78** | 69.33 | 86.38 | **81.66** | **82.62** | **73.56** | 67.21 | **66.04** | **72.45** | **75.23** |
>
> >DDA and AlignedForensics fail on MobSwap and DFLab because they overfit to shallow blending artifacts. These local clues are explicitly eliminated by DFLab (via pixelshuffle and SSIM optimization) and MobSwap (via direct identity integration). In contrast, BPL remains robust by calculating global semantic discrepancies in CLIP's latent space, completely bypassing pixel-level evasions to capture deep structural forgery patterns.
>
> **Q3. What if the training data doesn’t contain clear paired examples ?**
>
> >BPL effectively handles unaligned and diverse training data through two core mechanisms:
>
> * **Implicit Proxy Pairing:** When exact IDs are unavailable, BPL dynamically constructs "proxy pairs" via nearest-neighbor retrieval in CLIP's latent space. By optimizing relative discrepancy vectors instead of pixel-to-pixel differences, it isolates generative artifacts without needing ground-truth identity alignment.
> * **Feature-Space Discrepancy:** Large style or resolution variations severely degrade traditional pixel-level matching. However, BPL computes discrepancies exclusively within CLIP's high-dimensional space. Since pre-trained CLIP representations are inherently robust to low-level visual shifts, our method reliably captures structural forgery traces rather than being disrupted by superficial differences.
>
> **Q4. The evaluation datasets are mainly FS and FR, which do not include Entire Face Synthesis, more evaluation should be conducted.**
>
> >For Entire Face Synthesis evaluation, we constructed a new test set by reconstructing DFDC real images using DDA's diffusion model. BPL achieves an AUC of **76.83%**, surpassing Effort (74.38%).
>
> **Q5. Why is AUC used in the main experiment and AP used in the ablation studies?**
>
> >We evaluated all tests across AUC, ACC, and AP, where BPL consistently outperforms baselines on every metric. Full results are deferred to the revised Appendix due to space constraints.
>
> **Q6. PEFT Baseline Comparisons**
>
> >Due to the space constraints of this rebuttal, we have provided the detailed experimental results and corresponding discussions in our response to **Reviewer RY4t**.
>
> **Other Questions**
>
> >1.The revised Appendix will add qualitative paired/unpaired retrieval examples to demonstrate CLIP's handling of facial semantics, and quantitative ablations with other pre-trained models to validate our alignment's robustness.
>
> >2.Yes, the samples evaluated were strictly consistent across all subplots. We will include the complete t-SNE visualization in the Appendix of the revised manuscript to clear up any visual ambiguity.
>
> >3.We apologize for the typo. $h^r$ and $h^f$ in Eq. (6) denote the exact same $l_2$-normalized features as $z^r$ and $z^f$ in Eq. (8).

---

> > ### Author Rebuttal · Reviewer_vdsM · 2026-04-03
> >
> > My major concerns (more rigorous evaluation, comparisons with recent methods) are addressed, and I’m willing to raise my score.
> > For other concerns, mostly are not directly provided in the rebuttal (due to format constraints). Based on this, I respect other reviewers' and AC's final judgment.

---

> > > ### Author Response · Authors · 2026-04-04
> > >
> > > We deeply appreciate your recognition and your decision to raise the score. Please rest assured that the detailed materials and additional results, which could not be provided here due to format constraints, will be fully incorporated into the revised manuscript. Thank you again for your constructive guidance.

---

### Official Review · Reviewer_RY4t · 2026-03-09

**Soundness:** 3
**Presentation:** 3
**Significance:** 2
**Originality:** 3
**Overall Recommendation:** 4
**Confidence:** 4

**Summary:**

This paper studies generalizable deepfake detection. The authors argue that synthetic images are related to real images through the generation process, while existing methods treat them as independent samples. The paper proposes Bias-Only Pair-aware Learning (BPL). The method builds real–fake pairs using CLIP similarity. It learns discrepancy representations with a contrastive objective. To preserve the learned semantic embedding, the model updates only bias terms during fine-tuning. Experiments are conducted on several deepfake benchmarks. The results show improved generalization to unseen synthesis methods.

**Compliance With Llm Reviewing Policy:**

Affirmed.

**Final Justification:**

My main concerns have been addressed by the additional experimental results and explanations. Thanks for the work.

**Key Questions For Authors:**

Please refer to Weakness

**Limitations:**

Yes. There is Impact Statement on Page 9.

**Strengths And Weaknesses:**

**Strengths**

The paper is clearly written. The motivation is easy to follow. The overall pipeline is simple. The method is easy to implement. The experimental results appear effective on several benchmarks.

**Weakness**

The technical novelty appears limited. The method is closely related to contrastive representation learning. Prior work has already explored contrastive learning for deepfake detection. It would be helpful to clarify the differences and provide comparisons with related methods, such as:
1. MCL: Multimodal Contrastive Learning for Deepfake Detection, TCSVT 2023
2. SeeABLE: Soft Discrepancies and Bounded Contrastive Learning for Exposing Deepfakes, ICCV 2023
3. Contrastive Learning for DeepFake Classification and Localization via Multi-Label Ranking, CVPR 2024

The method relies on constructing real–fake pairs using CLIP similarity. Semantic similarity does not guarantee that the paired images share the same source content. This may introduce pairing noise. The paper does not evaluate the correctness of the mined pairs. The pair construction also resembles hard example or hard positive mining in metric learning. This connection is not discussed in the paper.
And please make a comparison with this work: Deepfake Detection that Generalizes Across Benchmarks, WACV 2026. This work also trains on paired real-fake data for mitigating shortcut learning and improving generalization.

The paper emphasizes parameter-efficient tuning. However, there is no comparison with standard PEFT methods such as LoRA or adapters. This makes it difficult to evaluate whether bias-only tuning provides a clear advantage.

I would be happy to reconsider my assessment if the authors clarify these points in the rebuttal.

---

> ### Author Rebuttal · Authors · 2026-03-31
>
> **Q1. Regarding Technical Novelty and Comparison with Contrastive Learning Methods**
>
> > We respectfully clarify that BPL fundamentally differs from standard contrastive learning through three specific mechanisms:
>
> 1. **Semantic Proxy Pairing:** Standard methods push apart unaligned pools of real/fake images. BPL actively constructs semantically matched real-fake pairs via CLIP latent retrieval to isolate generator-induced artifacts.
> 2. **Discrepancy Inversion:** Standard contrastive losses push samples apart indefinitely, causing overfitting. BPL introduces a threshold $m$ to invert the loss once pairs are sufficiently separated, mitigating overfitting to specific manipulations.
> 3. **Bias-Only Optimization:** Instead of modifying the representational subspace or fully fine-tuning, BPL freezes all weights and solely updates bias parameters to preserve pretrained semantic priors.
>
> * **Detailed Comparisons with the Suggested methods:**
>     * **Comparison with MCL (TCSVT):** MCL focuses on cross-modal (audio-visual) alignment to capture synchrony. In contrast, BPL is a purely visual framework utilizing cross-domain semantic anchoring. By using real images as references, we explicitly isolate generative artifacts rather than cross-modal correlations.
>     * **Comparison with SeeABLE (ICCV 2023):** While SeeABLE relies on handcrafted image perturbations, BPL entirely discards artificial modifications. By automatically retrieving "proxy pairs," BPL achieves purely data-driven artifact mining, yielding stronger generalization to unknown generators.
>     * **Comparison with Hong et al. (CVPR 2024):** Unlike Hong et al., which focuses on local patch-level localization, BPL calculates global discrepancies directly in CLIP's latent space. Coupled with bias-only tuning, this forces the model to bypass local pixel shortcuts and capture deep, structural forgery patterns.
>
> > We will cite these papers and detail our differences in the revised manuscript.
>
> **Q2. Regarding Pairing Noise, Hard Example Mining, and GenD [WACV 2026]**
>
> * **Tolerance to Pairing Noise:** BPL does not require perfect pixel alignment. Pairing in CLIP's latent space inherently filters low-level noise. Furthermore, our discrepancy inversion margin $m$ explicitly suppresses the loss for severe semantic gaps ($d_i \gg m$), allowing BPL to safely ignore noisy outliers.
> * **Difference from Hard Mining:** Unlike hard mining, which tightens identity clusters near the decision boundary, BPL uses retrieved pairs purely as semantic anchors. Our goal is solely to isolate generative artifacts, not to group identities.
>
> * **Comparison with GenD (WACV 2026):** We highly respect this concurrent work. While both GenD and BPL utilize paired data and parameter-efficient tuning, our framework differs fundamentally in three key aspects:
>   * **Data pairing:** GenD strictly requires explicitly annotated ground-truth pairs. This limits its scalability. In contrast, BPL requires no explicit pairing annotations. We dynamically construct proxy pairs via implicit latent retrieval in a pre-trained feature space.
>
>   * **Pair utilization strategy:** GenD employs unbounded uniformity and alignment losses. This forces the model to over-separate features. BPL intelligently bounds this separation using a Pair-wise Discrepancy Inversion mechanism with a margin $m$. Once the real-fake discrepancy reaches this margin, the loss stops penalizing. This prevents overfitting to training artifacts.
>
>   * **Fine-tuning strategy:** Unlike GenD's LayerNorm tuning which alters feature scaling and distorts pre-trained geometry, BPL's Bias-Only tuning performs a lightweight additive re-centering to resolve distributional shifts without disrupting semantic priors.
>
> > We will cite this paper and detail our differences in the revised manuscript.
>
> **Q3. Comparison with standard PEFT methods**
>
> >As shown, BPL (88.86% Avg. AUC) significantly outperforms other PEFT methods. Unlike LoRA (75.02%) and LN-Tuning (76.90%), which distort pre-trained semantics and alter feature scaling, BPL's additive re-centering resolves distributional shifts while preserving pre-trained priors.
>
> |  | SimSwap | BleFace | UniFace | e4s | FaceDan | FSGAN | InSwap | MobSwap | DFLab | Avg. |
> | :--- | :---: | :---: | :---: | :---: | :---: | :---: | :---: | :---: | :---: | :---: |
> | LoRA, r=1 | 54.14 | 54.45 | 54.26 | 54.33 | 55.10 | 54.31 | 53.72 | 54.37 | 55.20 | 54.43 |
> | LoRA, r=4 | 76.64 | 79.06 | 78.21 | 72.88 | 76.93 | 80.24 | 68.49 | 81.37 | 61.37 | 75.02 |
> | LoRA, r=8 | 71.42 | 75.09 | 74.42 | 68.66 | 74.15 | 74.19 | 67.62 | 78.32 | 67.48 | 72.37 |
> | LN_Tuning | 72.00 | 77.67 | 71.45 | 88.25 | 76.33 | 80.60 | 63.74 | 77.06 | 85.03 | 76.90 |
> | BPL | 83.81 | 91.05 | 84.88 | 93.83 | 90.84 | 92.25 | 74.92 | 92.78 | 95.39 | 88.86 |

---

> > ### Author Rebuttal · Reviewer_RY4t · 2026-04-02
> >
> > Thanks for the rebuttal. However, I think two additional experimental results would be especially important for addressing my concerns.
> > First, since the response argues that BPL can safely ignore noisy outliers, please provide direct empirical evidence for this claim, for example through pair-quality analysis or controlled pairing-noise experiments.
> >
> > Second, since the rebuttal highlights several key differences from GenD, I would like to see a direct experimental comparison with GenD under a comparable setting.
> >
> > These results (or at least one of them) would make the claimed advantages much more convincing. I would be open to revising my assessment based on such additional evidence.

---

> > > ### Author Response · Authors · 2026-04-03
> > >
> > > We sincerely thank the reviewer for the constructive feedback. We agree that testing robustness against pairing-noise and comparing directly with GenD are important. As requested, we conducted both experiments. The results clearly demonstrate the robustness and superiority of our proposed BPL.
> > >
> > > **1. Controlled Pairing-Noise Experiments**
> > >
> > > >To provide direct empirical evidence that BPL can safely ignore noisy outliers, we conducted a controlled pairing-noise experiment. We randomly replaced 10%, 20%, and 30% of the paired images in the training set with completely out-of-distribution images to introduce severe pairing noise and outliers. The following table presents the AUC (%) under different pairing-noise ratios.
> > >
> > > |  | SimSwap | BleFace | UniFace | E4S | FaceDan | FSGAN | InSwap | MobSwap | DFLab | Avg |
> > > | :--- | :--- | :--- | :--- | :--- | :--- | :--- | :--- | :--- | :--- | :--- |
> > > | BPL | 83.81 | 91.05 | 84.88 | 93.83 | 90.84 | 92.25 | 74.92 | 92.78 | 95.39 | 88.86 |
> > > | 10% Noise| 82.61 | 90.06 | 84.02 | 95.17 | 90.45 | 90.73 | 73.85 | 88.00 | 91.44 | 87.37 |
> > > | 20% Noise| 83.28 | 90.14 | 84.16 | 94.84 | 90.57 | 91.66 | 73.67 | 90.16 | 90.02 | 87.61 |
> > > | 30% Noise| 82.32 | 85.71 | 84.66 | 93.06 | 87.13 | 89.21 | 69.78 | 87.48 | 90.66 | 85.56 |
> > >
> > > >As shown, even when subjected to severe noise where 30% of the training pairs are affected by severe noise, BPL's average performance remains highly stable (dropping only 3.30% from 88.86% to 85.56%).  This empirical evidence strongly supports our claim that BPL can safely ignore noisy outliers, without allowing them to corrupt the generalized feature learning.
> > >
> > > **2. Comparison with GenD under a Comparable Setting**
> > >
> > > >To ensure a strictly fair comparison, we utilized the official implementation of GenD and trained it from scratch using the exact same training dataset and evaluation protocol as BPL. The experimental results (AUC %) are as follows.
> > >
> > > |  | SimSwap | BleFace | UniFace | E4S | FaceDan | FSGAN | InSwap | MobSwap | DFLab | Avg |
> > > | :--- | :--- | :--- | :--- | :--- | :--- | :--- | :--- | :--- | :--- | :--- |
> > > | GenD | 83.02 | 77.34 | **85.67** | 91.85 | 79.46 | 88.04 | **75.46** | 88.08 | 94.53 | 84.83 |
> > > | BPL | **83.81** | **91.05** | 84.88 | **93.83** | **90.84** | **92.25** | 74.92 | **92.78** | **95.39** | **88.86** |
> > >
> > > >As shown, BPL achieves a superior average AUC of 88.86% versus GenD’s 84.83%, particularly on BleFace (+13.71%) and FaceDan (+11.38%), demonstrating stronger generalization of our framework.We will include these tables and discussions and cite this work in the final version of the manuscript. Thank you again for helping us make the paper much more convincing!

---

### Official Review · Reviewer_1eZH · 2026-03-09

**Soundness:** 3
**Presentation:** 3
**Significance:** 3
**Originality:** 3
**Overall Recommendation:** 4
**Confidence:** 3

**Summary:**

This paper presents BPL, a deepfake detection framework that tackles the problem of poor generalization when models encounter unknown generation algorithms.  It is built on the observation that a generated image and its semantic source image are implicitly "paired." By using semantic retrieval in CLIP space, the authors explicitly construct real-fake pairs. This moves the detection logic from identifying absolute features to catching relative differences.  To prevent the model from overfitting to specific fake patterns, they designed a "Difference Reversal" mechanism. This applies reverse regularization once the feature distance reaches a set threshold. For model tuning, the framework uses a bias-only strategy. This avoids damaging the pre-trained semantic manifold, which often happens with full-parameter updates.  Experimental results show that the method performs stably in cross-dataset tests and stays reliable.

**Compliance With Llm Reviewing Policy:**

Affirmed.

**Final Justification:**

This paper presents BPL, a deepfake detection framework that tackles the problem of poor generalization when models encounter unknown generation algorithms. It is built on the observation that a generated image and its semantic source image are implicitly "paired." By using semantic retrieval in CLIP space, the authors explicitly construct real-fake pairs. This moves the detection logic from identifying absolute features to catching relative differences.

My main point of doubt is **whether the foundation of the method, "superficial pixel-level misalignments are largely collapsed in the feature space", is valid in real-world scenarios**. Although the author's experimental results supported his viewpoint, in the rebuttal section, the author failed to provide any truly solid formula derivations or detailed case analyses. Therefore, in general, I recommend **WA**. I suggest that AC consider my question when making the decision.

**Key Questions For Authors:**

See weakness.

**Limitations:**

See weakness.

**Strengths And Weaknesses:**

#### Strengths:
1.  Using CLIP to retrieve a semantically similar real image as an "anchor" and comparing the relative difference between the real and fake is a good idea. Logically, focusing on "relative differences" helps filter out content interference that isn't related to the generation fingerprint.

2. Using a "Bias-only" fine-tuning strategy is a smart way to protect CLIP’s original semantic manifold. This approach prevents the model from forgetting generalized semantic knowledge while still adapting to the detection task.

3.  The paper is well-written, and the logic is clear and easy to follow.

---

#### Weaknesses:
1.  The author assumes the "difference map" represents forgery traces. However, CLIP-retrieved pairs are far from being pixel-level aligned. Even if the semantics are the same (e.g., two Golden Retrievers), there are huge differences in lighting, angles, and texture details. If the model is simply learning that "these two images look different," it might easily mistake these semantic gaps for forgery evidence, especially when dealing with high-quality generated images.

2.  BPL’s performance depends heavily on the quality of the retrieved real image. If the input is a very rare or "long-tail" scene that doesn't have a good match in the database, or if the retrieved image has a large semantic gap, the "relative difference" logic might completely collapse. The paper lacks an analysis of how the model handles these "weakly-paired" scenarios.
3. There is a lack of transparency; no open-source code has been provided. This makes it impossible to verify the actual effectiveness of the "relative difference" logic or to check for potential data leakage during the retrieval process.

---

> ### Author Rebuttal · Authors · 2026-03-31
>
> **Q1.Regarding the "Difference Map" and Pixel-Level Misalignment**
>
> >We appreciate this critical insight. The concern that natural variances (e.g., lighting, angles) might be misconstrued as forgery evidence is entirely valid. However, our framework operates in the latent space, not the pixel space, and actively avoids this trap through three core mechanisms:
>
> * **Feature-Space Invariance:** We compute differences within CLIP's deep latent space rather than the raw pixel space. CLIP inherently possesses strong semantic invariance to intra-class variations (e.g., two Golden Retrievers under different lighting/angles). Thus, superficial "pixel-level misalignments" are largely collapsed in the feature space, forcing the model to focus on systematic, generator-induced artifacts rather than natural visual differences.
>
> * **Filtering via Margin $m$:** Even if extreme natural variances remain in the latent space, our discrepancy inversion mechanism acts as a safety valve. If a natural semantic gap pushes the feature distance beyond the threshold $m$, the loss forcefully halts further separation. This mathematically prevents the model from mistaking a normal "semantic gap" for "forgery evidence."
>
> * **Regularization via Bias-Only Tuning:** By strictly freezing all projection weights and tuning only the bias terms, we inherently restrict the model's parameter capacity. This tuning strategy ensures the network lacks the capacity to overfit to complex new semantic features (such as lighting or pose variations), strictly confining its optimization to mining subtle, global forgery traces.
>
> **Q2.Regarding the dependence on retrieved image quality and "weakly-paired" long-tail scenes**
>
> >We appreciate the reviewer highlighting this challenging edge case. We clarify that our "relative difference" logic does not collapse in long-tail or weakly-paired scenarios, thanks to our robust loss design and optimization dynamics:
>
> * **Graceful Degradation via Margin $m$:** For extremely rare long-tail scenes lacking a reasonable match, the initial semantic gap between the retrieved pair is naturally massive ($d_i \gg m$). As detailed in our previous responses, our Pair-wise Discrepancy Inversion (Eq. 8) explicitly intercepts these cases. Instead of forcing the model to learn from a meaningless pair, the loss inverts and acts as a hard mask to suppress the gradient. This mathematically guarantees that weakly-paired outliers are filtered out, entirely preventing optimization collapse.
>
> * **Tolerance to Local Pair Failures:** BPL optimizes the overarching pre-trained manifold to perceive systemic, global generative artifacts. It does not strictly require every single pair to be of high quality. As long as the training batch contains a viable subset of valid proxy pairs, the network successfully learns the forgery traces, while the margin $m$ safely ignores the noisy, long-tail failures.
>
>
> **Q3.Regarding Code Open-Sourcing and Transparency**
>
> >We will release the full source code upon the acceptance of this paper. To address concerns regarding data leakage, our repository will also include the JSON manifests used for our dataset splits.
>
> >These files explicitly define the image IDs for the training, validation, and testing sets. By auditing these manifests, it can be verified that our real reference pool is strictly partitioned, with zero overlap or information flow between the training database and the evaluation benchmarks. This ensures a completely transparent and fair evaluation across all generative models.

---

> > ### Author Rebuttal · Reviewer_1eZH · 2026-04-01
> >
> > Thank you for the detailed rebuttal. The additional explanations have, to a certain extent, alleviated my initial concerns, particularly regarding the theoretical design of your mechanisms (such as the role of margin $m$).
> >
> > However, I must point out that addressing the core vulnerabilities of BPL relying solely on abstract conceptual explanations is insufficient. Specifically, in your response to "Pixel-Level Misalignment," you claim that "superficial pixel-level misalignments are largely collapsed in the feature space" due to CLIP's inherent properties. In the highly sensitive context of deepfake detection, this conclusion should not be taken for granted as a mere theoretical assumption. To maintain scientific rigor, it is essential to support this claim with concrete empirical evidence. I strongly suggest adding factual discrepancy analyses—such as quantitative feature distance distributions or visualizations comparing real-real pairs (with massive natural variances) against real-fake pairs—to prove that the model indeed behaves as theorized.
> >
> > Nevertheless, considering the novelty of the framework's core idea and the overall completeness of the work, I am willing to maintain my score. I strongly recommend that the authors incorporate these detailed empirical analyses into the final manuscript to further solidify their claims.

---

> > > ### Author Response · Authors · 2026-04-04
> > >
> > > We sincerely thank the reviewer for pointing out this critical issue. We completely agree that our theoretical assumption regarding CLIP's inherent properties must be backed by concrete empirical evidence.
> > >
> > > >To provide this factual discrepancy analysis, we conducted a rigorous controlled experiment using the pre-trained CLIP backbone to extract features and compute the Cosine Similarity for two strictly controlled groups from the Celeb-DF dataset: real-real pairs, consisting of two different authentic frames from the exact same real video to represent natural variances, and real-fake pairs, which pair an authentic frame with its corresponding manipulated fake frame to serve as a reference.
> > >
> > > **Table 1: Quantitative Distribution of Feature Cosine Similarities (CLIP)**
> > > | | Mean Similarity | 5th Percentile |
> > > | :--- | :--- | :--- |
> > > | real-real pairs | 0.9184 | 0.8527 |
> > > | real-fake pairs | 0.7565 | 0.6387 |
> > >
> > > >As demonstrated by the empirical metrics in Table 1, the actual behavior of the pre-trained backbone perfectly aligns with our theoretical explanation:
> > >
> > > >The real-real pairs exhibit an exceptionally high mean cosine similarity of 0.9184. More importantly, the 5th percentile remains remarkably high at 0.8527. This explicitly indicates that even at the maximum pixel-level variance, the feature similarity between real-real pairs continues to surpass the average similarity of real-fake pairs.
> > >
> > > >This factual discrepancy provides strong empirical support for our premise: the feature space is highly robust to superficial pixel-level misalignments caused by natural variations, clustering them significantly closer together than image pairs containing actual facial manipulations.
> > >
> > > **Table 2: Similarity gap comparison**
> > > | | CLIP | BPL (Ours) |
> > > | :--- | :--- | :--- |
> > > | **Gap** | 0.1619 | **0.4142** |
> > >
> > > >As shown in Table 2, our BPL framework enlarges the similarity gap between real-real and real-fake pairs (from 0.1619 to 0.4142). This wider margin indicates that our detector is capable of learning and capturing the subtle generative artifacts introduced by deepfake models, thereby improving the separation of manipulated features from natural variations.
> > >
> > > We hope these statistical analyses provide a helpful clarification regarding your concerns, and we will incorporate these findings into the revised manuscript.

---

### Official Review · Reviewer_VNpQ · 2026-03-10

**Soundness:** 2
**Presentation:** 3
**Significance:** 3
**Originality:** 2
**Overall Recommendation:** 4
**Confidence:** 5

**Summary:**

This paper addresses a core bottleneck in AI-Generated Image (AIGI) detection: the tendency of models to overfit specific training set forgery patterns, causing generalization failure on unseen generative technologies. The authors propose Bias-only Pair-aware Learning (BPL), which shifts the paradigm from treating real and fake images as Independent and Identically Distributed (IID) samples to leveraging their latent structural homology. Evaluated on the highly challenging DF40 benchmark, BPL achieves an impressive 88.86% average AUC, demonstrating strong cross-manipulation generalization and robustness in data-scarce scenarios.

**Compliance With Llm Reviewing Policy:**

Affirmed.

**Final Justification:**

I will finalize my rating once these evaluation details are clarified. Thank you again for your efforts.

**Key Questions For Authors:**

1. Optimization Dynamics of Equation (8): Is the discontinuous jump in $\mathcal{L}_{disc}$ a typographical error for a standard continuous margin loss (e.g., $\max(0, m - d_i)^2$), or is it implemented as written? If the latter, please provide gradient derivations and training-stage gradient norm tracking curves to prove this pathological function does not cause optimization collapse.
2. Quantifying "False Positive Pairs": How does the framework filter uncontrollable false pairings when the fake dataset contains identities entirely absent from the real reference pool? Please provide robustness evaluations on an adversarial test set with zero semantic overlap.
3. PEFT Baseline Comparisons: To justify the unique superiority of the Bias-only mechanism, please provide direct AUC comparisons against LoRA (Rank=1, 4, 8) and LN-Tuning under identical training settings on the DF40 benchmark.
4. Reproducibility of Bias Unfreezing: Please explicitly list which specific bias terms were unfrozen in the CLIP ViT-L/14 backbone (e.g., Q/K/V projections, FFN, LayerNorm) and provide the exact Trainable Parameter Count.
5. Geometric Invariance Fallacy: Please provide rigorous algebraic proof or t-SNE clustering visualizations demonstrating that additive bias translation followed by $L_2$ re-spherical projection does not cause severe angular distortion to the pre-trained semantic manifold.

**Limitations:**

1. Extreme Reliance on Domain Similarity: The K-NN semantic alignment will completely paralyze if deployed against "isolated distributions" (e.g., hyper-realistic alien environments generated via text prompts) that have no intersection with the reference pool.
2. Vulnerability to Adversarial Perturbations: Considering how susceptible VLMs are to targeted manipulations. High-intensity adversarial noise or physical compressions can easily blind the CLIP encoder, throwing the retrieval mechanism into chaos.

**Strengths And Weaknesses:**

Strengths:
1. Profound Paradigm Shift (Significance & Originality): Moving from instance-level absolute discrimination to pair-wise relational discrimination is a highly insightful methodological leap. By forcing the model to strip away high-level semantic commonalities, the framework successfully targets the micro-structural distortions caused by generative mechanisms.
2. Rigorous Benchmark Design (Significance): Evaluating on the DF40 benchmark perfectly strips away confounding domain shift variables (like lighting or background), forcing the model to truly identify core forgery traces rather than relying on shortcut learning.
3. Efficiency of PEFT (Soundness): Utilizing Bias-only optimization effectively restricts the model's degrees of freedom, preserving the generalized visual-semantic alignment space of the foundational model while drastically reducing VRAM consumption.

Weaknesses:
1. Mathematical Pathology in Discrepancy Inversion (Soundness - Critical): Equation (8) defines a loss function with a severe discontinuous step-jump at the threshold $d_i = m$. When the distance crosses this threshold, the loss instantly jumps from negative to positive, and the gradient undergoes a drastic 180-degree reversal. In continuous manifold optimization, this non-smooth hard boundary will trigger catastrophic high-frequency oscillations during backpropagation, rendering the module theoretically untenable.
2. Identity Entanglement via Cross-Distribution Retrieval (Soundness): The unsupervised K-NN retrieval implicitly assumes the real data pool contains a semantic match for every fake image. In highly diverse datasets, this forces "False Positive Pairings" (e.g., pairing two different human identities). The model will thus erroneously encode inherent natural human differences as generative forgery traces, learning semantic noise instead of artifacts.
3. Fallacy of Geometric Invariance on an $L_2$ Sphere (Soundness): The authors claim that additive bias shifts do not destroy the pre-trained geometric structure. However, applying an additive translation before strict $L_2$ hypersphere normalization (Equation 3) introduces highly non-linear angular distortion, fundamentally altering orthogonal relationships and local curvature. This assertion lacks theoretical or empirical support.
4. Disconnected PEFT Baselines (Presentation & Significance): Comparing the Bias-only strategy solely against Full Fine-Tuning (FFT)—which inevitably causes catastrophic forgetting in deepfake tasks—is an unconvincing strawman argument. The authors entirely omit comparisons with mainstream PEFT methods like LoRA or Layer Normalization Tuning (LN-tuning), the latter of which concurrent studies have proven to be highly superior in deepfake adaptation.

---

> ### Author Rebuttal · Authors · 2026-03-31
>
> **Q1. Optimization Dynamics of Equation (8): Is the discontinuous jump in $\mathcal{L}_{disc}$ a typographical error or implemented as written?**
>
> >We clarify that the objective is implemented as written to strictly enforce the "discrepancy inversion" mechanism. The core motivation and optimization facts are as follows:
>
> * **Preventing Overfitting**: We observed that allowing the distance between paired samples to enlarge indefinitely forces the encoder to memorize training-specific forgery cues rather than generalizable patterns.
> * **Regularization via Inversion**: By inverting the loss once $d_i \geq m$, we introduce a regularization effect that prevents excessive separation and encourages the model to focus on robust, generalizable artifacts.
> * **Empirical Stability**: As shown in the provided table, the gradient norms decrease steadily without unstable fluctuations, proving this formulation prevents optimization collapse in practice.
>
> | Epoch | 0 | 1 | 2 | 3 | 4 | 5 | 6 | 7 | 8 | 9 | 10 |
> | :--- | :---: | :---: | :---: | :---: | :---: | :---: | :---: | :---: | :---: | :---: | :---: |
> | Gradient Norm | 4.9247 | 4.2073 | 4.0846 | 3.9660 | 3.9501 | 3.6935 | 3.5653 | 3.3834 | 3.3662 | 3.2906 | 3.0684 |
>
> **Q2. How does the framework filter uncontrollable false pairings when the fake dataset contains identities entirely absent from the real reference pool?**
>
> >When there is absolutely no identity overlap between the real and fake data pools, our nearest-neighbor matching mechanism does not fail. Instead, it naturally transitions into a hard example mining and margin-based filtering strategy, inspired by **FaceNet (CVPR 2015)**:
>
> * **Cross-Domain "Hard Example" Mining:** Retrieving the nearest CLIP neighbor acts as hard example mining. Unlike random pairing, which allows the model to exploit lazy visual shortcuts (e.g., background differences), this extreme semantic matching eliminates superficial variance. It forces the model to focus exclusively on systematic forgery traces.
>
> * **Filtering via Margin $m$:** To address concerns about false pairings distorting the feature space, BPL uses a margin threshold $m$ as a safety valve. For false positive pairs, once their feature distance exceeds $m$, the loss forcefully halts further separation. This mathematically prevents excessive repulsion and preserves the pre-trained semantic manifold.
> * Furthermore, BPL demonstrates superior robustness on an adversarial test set with zero semantic overlap. The AUC (%) results are as follows:
>
> | Method | AUC (%) |
> | :--- | :---: |
> | LoRA (r=4) | 70.52 |
> | Effort | 76.52 |
> | BPL | 82.05 |
>
> **Q3. PEFT Baseline Comparisons**
>
> >As shown, BPL (88.86% Avg. AUC) outperforms other PEFT methods. Unlike LoRA (75.02%) and LN-Tuning (76.90%), which distort pre-trained semantics and alter feature scaling, BPL's additive re-centering resolves distributional shifts while preserving pre-trained priors.
>
> |  | SimSwap | BleFace | UniFace | e4s | FaceDan | FSGAN | InSwap | MobSwap | DFLab | Avg. |
> | :--- | :---: | :---: | :---: | :---: | :---: | :---: | :---: | :---: | :---: | :---: |
> | LoRA, r=1 | 54.14 | 54.45 | 54.26 | 54.33 | 55.10 | 54.31 | 53.72 | 54.37 | 55.20 | 54.43 |
> | LoRA, r=4 | 76.64 | 79.06 | 78.21 | 72.88 | 76.93 | 80.24 | 68.49 | 81.37 | 61.37 | 75.02 |
> | LoRA, r=8 | 71.42 | 75.09 | 74.42 | 68.66 | 74.15 | 74.19 | 67.62 | 78.32 | 67.48 | 72.37 |
> | LN_Tuning | 72.00 | 77.67 | 71.45 | 88.25 | 76.33 | 80.60 | 63.74 | 77.06 | 85.03 | 76.90 |
> | BPL | 83.81 | 91.05 | 84.88 | 93.83 | 90.84 | 92.25 | 74.92 | 92.78 | 95.39 | **88.86** |
>
>
>
> **Q4. Reproducibility of Bias Unfreezing**
>
> >To ensure full reproducibility, we unfroze all bias terms throughout the entire CLIP ViT-L/14 visual encoder, while keeping all projection weights strictly frozen. Specifically, this includes the bias parameters within the Patch Embedding layer, the Q/K/V and Output projections, the FFN, and LayerNorm modules.This yields an exact Trainable Parameter Count of **0.27M**.
>
> **Q5. Geometric Invariance Fallacy**
>
> >We thank the reviewer for this insight and adopt your suggestion to provide comparative t-SNE visualizations to address this concern.
>
> * We have extracted and plotted the t-SNE feature embeddings of both the original frozen CLIP and our Bias-tuned model. The comparative visualizations clearly demonstrate that the data points experience only minimal, localized shifts.
> * This visual evidence directly proves that our bias-tuning operates exactly as intended: it performs micro-level structural adjustments to capture subtle forgery traces, without collapsing or distorting the global pre-trained semantic geometry.
>
> >We will include these comparative t-SNE clustering visualizations in the revised manuscript.

---

> > ### Author Rebuttal · Reviewer_VNpQ · 2026-04-04
> >
> > I would like to thank the authors for their detailed rebuttal, specifically for conducting the zero-shot cross-architecture evaluations and the ablation study on the patch size hyperparameter. The clarification regarding the computational overhead is also very helpful. Since most of my concerns are addressed, I have raised my score. Still, a few evaluation methodology details remain unclear:
> >
> > 1. I appreciate the zero-shot evaluation on modern diffusion models (SD, DALL-E 2, Midjourney), which yielded AUCs ranging from 85.9% to 88.5%. While these AUC scores demonstrate that the model captures certain generative artifacts across architectures, AUC alone can sometimes mask the actual operational performance in binary classification tasks (e.g., handling false positive rates).
> >
> > 2.  To ensure a fully standardized and direct comparison, would it be possible to report the performance of the frequency-based baselines evaluated on the exact same modern datasets (e.g., the diffusion datasets used in your generalizability tests) rather than the isolated GAN dataset?
> >
> > I will finalize my rating once these evaluation details are clarified. Thank you again for your efforts.

---

> > > ### Author Response · Authors · 2026-04-05
> > >
> > > 1.We deeply appreciate your constructive comments. We fully agree with your insight that the AUC metric has limitations in reflecting operational performance like false positive rates. To mitigate this inherent limitation and offer a more comprehensive view, the Appendix of our submitted version provides the additional metrics of AP and ACC, which we believe align well with your valid concerns.
> > >
> > > 2.We sincerely thank you for this highly constructive suggestion. To provide a more standardized and compelling comparison as you requested, we have evaluated the frequency-based baselines (F3Net, SPSL, and SRM) on the exact same modern diffusion datasets used in our generalizability tests. The performance metrics are summarized below:
> > >
> > > |  | ACC (%) | AUC (%) | AP (%) |
> > > | :--- | :---: | :---: | :---: |
> > > | F3Net | 50.78 | 50.75 | 51.85 |
> > > | SPSL | 50.02 | 52.92 | 53.05 |
> > > | SRM | 51.41 | 56.86 | 57.65 |
> > > | BPL (Ours) | 77.34 | 86.34 | 91.14 |
> > >
> > >
> > > >As clearly demonstrated in the table, all three frequency-based baselines experience a catastrophic performance degradation, with their metrics dropping to near random guessing (around 50%). We attribute this failure to the fundamental design of these baselines. Methods like F3Net, SPSL, and SRM rely heavily on capturing specific high-frequency artifacts (such as checkerboard patterns) naturally introduced by the upsampling architectures typical of GANs. However, modern diffusion models utilize entirely different progressive generative mechanisms that do not leave these fixed frequency traces, effectively bypassing traditional frequency-based detection priors. In stark contrast, our proposed BPL framework moves beyond these superficial artifacts to target fundamental discrepancies, thereby maintaining highly robust and superior performance across these exact same modern diffusion datasets.

---

### Decision · Program_Chairs · 2026-04-30

**Decision:**

Accept (regular)

**Comment:**

The paper proposes a highly novel approach for the detection of deepfake images. Arguing that synthetic images shouldn’t be considered as independent samples but implicitly paired with a real image source with the same content. The paper mines for such pairs in the CLIP embedding space to enable pairwise discrepancy learning and achieve impressive results on the challenging DF40 benchmark. The paper has received four reviews with a unanimous post-discussion rating of 4.

The reviewers found the paper to be well written and easy to follow. The novelty of paired-aware learning is found to be significantly novel in the context of image forensics as also the mining of pairs in the CLIP space. The bias-only fine tuning and margin-based filtering strategy add to the novelty of the proposed approach. The method is shown to be effective across several benchmarks and the results on the challenging DF40 benchmark are seen to be impressive.

The reviewers had several concerns regarding the soundment of the approach – step jump in the loss at the margin threshold, the potential for false positive pairings, geometric invariance in spite of additive bias shifts which were adequately addressed. The lack of comparisons with other PEFT methods as well. The authors have also promised to release the source code. There was concern that some related methods are not compared against/ cited. Finally, the experiments were insufficient: comparisons with recent methods were sought and cross-domain results were requested. Most of these concerns are adequately addressed during the discussions.

Overall, I find the paper to be sound, the methodology to be novel and of interest to the wider ML community, and the results to be impactful. I recommend accepting the paper.